# Organismal and Cellular Stress Responses upon Disruption of Mitochondrial Lonp1 Protease

**DOI:** 10.3390/cells11081363

**Published:** 2022-04-16

**Authors:** Eirini Taouktsi, Eleni Kyriakou, Stefanos Smyrniotis, Fivos Borbolis, Labrina Bondi, Socratis Avgeris, Efstathios Trigazis, Stamatis Rigas, Gerassimos E. Voutsinas, Popi Syntichaki

**Affiliations:** 1Laboratory of Molecular Genetics of Aging, Biomedical Research Foundation of the Academy of Athens, Center of Basic Research, 11527 Athens, Greece; eirinitaouktsi@gmail.com (E.T.); ekyriakou@bioacademy.gr (E.K.); fborbolis@gmail.com (F.B.); labrinabondi@gmail.com (L.B.); e.trigazis18@imperial.ac.uk (E.T.); 2Department of Biotechnology, Agricultural University of Athens, 11855 Athens, Greece; srigas@aua.gr; 3Laboratory of Molecular Carcinogenesis and Rare Disease Genetics, Institute of Biosciences and Applications, National Center for Scientific Research “Demokritos”, 15341 Athens, Greece; stefanos.smyrniotis@gmail.com (S.S.); savgeris@bio.demokritos.gr (S.A.)

**Keywords:** mitochondria, LonP1, aging, cancer, CDDO-Me, *C. elegans*

## Abstract

Cells engage complex surveillance mechanisms to maintain mitochondrial function and protein homeostasis. LonP1 protease is a key component of mitochondrial quality control and has been implicated in human malignancies and other pathological disorders. Here, we employed two experimental systems, the worm *Caenorhabditis elegans* and human cancer cells, to investigate and compare the effects of LONP-1/LonP1 deficiency at the molecular, cellular, and organismal levels. Deletion of the *lonp-1* gene in worms disturbed mitochondrial function, provoked reactive oxygen species accumulation, and impaired normal processes, such as growth, behavior, and lifespan. The viability of *lonp-1* mutants was dependent on the activity of the ATFS-1 transcription factor, and loss of LONP-1 evoked retrograde signaling that involved both the mitochondrial and cytoplasmic unfolded protein response (UPR^mt^ and UPR^cyt^) pathways and ensuing diverse organismal stress responses. Exposure of worms to triterpenoid CDDO-Me, an inhibitor of human LonP1, stimulated only UPR^cyt^ responses. In cancer cells, CDDO-Me induced key components of the integrated stress response (ISR), the UPR^mt^ and UPR^cyt^ pathways, and the redox machinery. However, genetic knockdown of LonP1 revealed a genotype-specific cellular response and induced apoptosis similar to CDDO-Me treatment. Overall, the mitochondrial dysfunction ensued by disruption of LonP1 elicits adaptive cytoprotective mechanisms that can inhibit cancer cell survival but diversely modulate organismal stress response and aging.

## 1. Introduction

Lon proteases belong to the highly evolutionarily conserved AAA^+^ (ATPases Associated with diverse cellular Activities) protease family and are present in all eukaryotic organelles, including mitochondria, chloroplasts, and peroxisomes [1,2]. Lon’s primary function is to prevent the aggregation of unfolded/misfolded or oxidized proteins in the organelles by acting as both chaperone and protease [3,4]. In yeast and mammalian cells, loss of mitochondrial Lon protease (hereafter termed LonP1) caused mitochondrial dysfunction, accumulation of damaged proteins, and cell death via apoptosis and necrosis, supporting its central role in protein quality surveillance and maintenance of cellular homeostasis [5,6,7,8,9]. LonP1 forms a homohexameric ring-shaped structure in humans and protects mitochondria against aggregation by solubilizing specifically newly imported proteins and degrading unprocessed, misfolded, and oxidized proteins in the mitochondrial matrix [10,11]. Additionally, LonP1 is implicated in a wide range of regulatory cellular processes, such as mitochondrial DNA maintenance, mitochondrial unfolded protein response, the metabolic adaptation of tumor cells, or stress adaptation of *Drosophila melanogaster* [12,13,14,15].

In humans, LonP1 mutations are associated with CODAS syndrome, a rare developmental disorder affecting multiple organs [16]. Moreover, in several types of human tumors or cancer cell lines, LonP1 levels were increased, suggesting a high level of unfolded protein (proteotoxic) stress within mitochondria [17]. Even though the function of LonP1 in tumorigenesis is still poorly understood, inhibition of the protease is likely associated with decreased rates of tumor cell growth, proliferation, and metastasis [18]. This evidence suggests that LonP1 may be a potential target for cancer therapy. In support of its role in proteostasis, the levels of LonP1 are induced during acute oxidative stress, heat shock, and hypoxia [19], whereas both the level and activity of LonP1 are decreased during chronic oxidative stress or aging [20,21]. Interestingly, deletion of mitochondrial Lon in the budding yeast *Saccharomyces cerevisiae* or the filamentous ascomycete *Podospora anserina* accelerated aging, linking LonP1 function to cellular senescence [22,23]. In *Drosophila melanogaster*, both RNAi interference (RNAi)-mediated knockdown and constitutive overexpression of LonP1 decreased normal lifespan, indicating that dysregulation of LonP1 levels has a negative impact on longevity [15]. However, the role of mitochondrial LonP1 protease (named LONP-1) in *C. elegans* aging or organismal adaptation to stress has not been established so far.

In *C. elegans*, mitochondrial dysfunction induces defense mechanisms and adaptive responses that could lead to enhanced longevity. The mitochondrial unfolded protein response (UPR^mt^) is a retrograde signaling pathway, well-studied in *C. elegans*, which is conserved among species and helps cells to adapt to mitochondrial stress [24,25,26]. Although UPR^mt^ is responsible for the long lifespan of certain mitochondrial mutants with impaired electron transport chain (ETC) activities or mitonuclear imbalance [27,28,29,30], UPR^mt^ activation can be uncoupled from longevity [31]. As a pivotal regulator of UPR^mt^, ATFS-1 (activating transcription factor associated with stress-1) translocates to the nucleus under mitochondrial stress and coordinates a broad alteration of gene expression to ensure proteostasis, detoxification, and metabolic reprogramming [32,33]. In the absence of stress, ATFS-1 acts as a bidirectional bZIP transcription factor entering the mitochondrial matrix, where it is degraded by LONP-1 protease [32]. Remarkably, RNAi-mediated depletion of LONP-1 resulted in the accumulation of ATFS-1 in mitochondria without inducing the UPR^mt^ pathway [32]. Additional important regulators of the UPR^mt^ in *C. elegans* have been identified to act in parallel or in concert with ATFS-1 [34,35,36,37,38].

In mammals, the UPR^mt^ is a conserved process that induces the expression of cytoprotective genes in response to stressed mitochondria [39]. The pathway involves the activating transcription factor 5 (ATF5), which is regulated similarly to ATFS-1, as it recapitulates the UPR^mt^ in *atfs-1* loss-of-function worms [40]. Further studies have revealed the importance of the integrated stress response (ISR) and the activating transcription factor 4 (ATF4) in mounting a transcriptional response to mitochondrial dysfunction [41,42]. Moreover, ATF4 and the nuclear factor erythroid 2-related factor 2 (NRF2), a major regulator of the cellular antioxidant defense mechanisms, can cooperatively regulate the expression of heme oxygenase 1 (HO-1) and the proapoptotic CCAAT/enhancer-binding protein homology protein (CHOP) during mitochondrial stress [43]. In human HeLa cells, depletion of LonP1 increased mitochondrial ROS production and the levels of oxidized proteins [9], though discrepancies still exist regarding the induction of stress-responsive signaling pathways, such as the UPR^mt^ [44] or the ISR pathways [41].

Herein, we have investigated the effects of LonP1 deficiency at the molecular, cellular, and organismal levels in two experimental systems, the nematode *C. elegans* and human cancer cells. Using both genetic and pharmacological techniques, we analyzed transcriptional responses to LonP1 inhibition and how they affect physiological processes and phenotypes. Mitochondrial perturbation by LonP1 deficiency seems to act via many specific cellular mechanisms to induce distinct stress responses best suited to the cells and organism. Elucidating these adaptive homeostatic mechanisms at both the cellular and the organismal/systemic levels is important in understanding how dysregulation of LonP1 may have a pathogenic impact leading to disease.

## 2. Materials and Methods

### 2.1. C. elegans Strains and Culture Conditions

Standard methods of culturing and handling worms were used [45]. Worms were raised on NGM plates seeded with *Escherichia coli* OP50 and supplemented, whenever it was deemed necessary, with 40 μg/mL 5-fluoro-2′-deoxyuridine (FUdR, Sigma-Aldrich, St. Louis, MO, USA) to prevent progeny growth. For CDDO-Me treatment, NGM plates with UV-killed bacteria supplemented with 5 μM and 10 µM 2-cyano-3,12-dioxo-oleana-1,9(11)-dien-28-oic acid methyl ester (Cayman Chemical, Ann Arbor, MI, USA), or equivalent amount of the solvent DMSO were used. All *C. elegans* strains used in this study are presented in Appendix A. The *lonp-1(ko)* mutant was generated by the CRISPR-Cas-mediated gene-editing method in N2 (wild-type) worms [46,47], and injections/screens were provided by Invermis Limited, London, UK. The crRNAs used to delete the complete open reading frame of LONP-1, as well as all primers used in this study, are listed in Appendix A. Double mutants were generated by crossing the corresponding strains and phenotypic or PCR-based selection in F2 progeny. Transgenic *lonp-1::gfp* animals were generated by microinjection of plasmid DNAs into the gonad of N2 young adults, using *rol-6(su1006)* as a cotransformation marker. Multiple lines were obtained and screened for the representative expression pattern.

### 2.2. Human Cell Cultures

Normal skin fibroblast (DSF22), primary melanoma (WM115), metastatic melanoma (WM266-4), and fibrosarcoma (HT1080) cell lines were cultured in 1× DMEM (Dulbecco’s modified Eagle’s medium, Biowest, Nuaillé, France) supplemented with 10% FBS (fetal bovine serum, Biowest, Nuaillé, France), and 1% Penicillin/Streptomycin (Gibco, Waltham, MA, USA) at 37 °C, 5% CO_2_ and ≥95% humidity. WM266-4 and HT1080 cancer cell lines were treated with 1 μΜ CDDO-Me for 24 h. All other cases in which different concentrations or time points were used are shown in the text.

### 2.3. RNA Interference

RNAi experiments in worms were performed on NGM plates seeded with *E. coli* HT115 (DE3) bacteria transformed with the indicated RNAi construct (primers listed in Appendix A) or the appropriate empty vector (plasmid L4440 or T444T purchased from Addgene, Watertown, MA, USA). All RNAi assays were performed at 20 °C, as previously described [48]. For human LonP1 small interfering RNA (siRNA) experiments, dicer-substrate RNA (DsiRNA) duplexes (5′-AAUCAGAGUGUGGCAUAGAAGCUAT-3′) were purchased from Integrated DNA Technologies (IDT, Coralville, IA, USA). A predesigned DsiRNA to be used as a negative control was also purchased from IDT. SiRNA transfections were performed using Lipofectamine2000 (Invitrogen, Waltham, MA, USA) according to the manufacturer’s protocol. Briefly, 2 × 10^5^ cells (WM266-4 or HT1080) were seeded in a 35 mm culture dish. The next day, 20 nM DsiRNA were transfected with 5 μL Lipofectamine2000 in Opti-MEM (Gibco, Waltham, MA, USA). After 48 h, cells were subcultured (ratio 1:6) and retransfected to a final incubation period of 144 h.

### 2.4. Phenotypic Analysis of Worms

Eggs were synchronized by egg-lay for a 2–3 h period on NGM plates with adequate food and allowed to hatch at 20 °C. The number of progenies in each developmental stage after 72 h was counted, and the percentage of population distribution across worm genotypes was calculated. Total brood size and egg-laying time were determined at 20 °C by picking single L4 larvae onto OP50 plates and transferring them daily to new plates until egg laying had stopped. The F1 progeny laid by each individual worm was scored on each plate after egg hatching. To examine locomotion, 1day adults were transferred to freshly prepared NGM plates with thin lawns at room temperature. Complete body bends (every time the part of the worm just behind the pharynx reached a maximum bend in the opposite direction from the bend last counted) were counted for each individual for 20 s. Similarly, 1-day adults were placed on a plate without bacteria to crawl freely for 30 s and then were transferred to a drop of M9 buffer, where the frequency of thrashing was estimated for 30 s, under a dissecting microscope. The body bends and thrashing rate of 1 min were plotted for each worm per strain. For food avoidance assays, approximately 100 eggs from each strain were transferred to the center of mating-like plates (with small circular lawns of uniform size) and incubated at 20 °C for 48–72 h. The percentage of “runaway” worms from the lawn was scored for each plate (N_off_/N_total_ × 100).

### 2.5. Lifespan Analysis

*C. elegans* lifespan analysis was conducted at 20 °C or 25 °C as described previously [49]. Briefly, 100 to 150 animals of each strain in late L4 larvae to a young adult stage were transferred to NGM plates (30–40 per plate) seeded with OP50 or HT115 (DE3) bacteria (Day 1 of lifespan assay) and moved to fresh plates every 2–4 days. When indicated, plates supplemented with 40 µM FUdR were used. Viability was scored daily, and worms that failed to respond to stimulation by touch were considered dead. Bagged or raptured worms and animals that crawled off the plates are referred to as censored in the analysis. Statistical analysis was performed by comparing each population to the appropriate control, and *p* values were determined using the log-rank (Mantel–Cox) test. Replicates were carried out as indicated in Appendix A.

### 2.6. Microscopic Analysis

Synchronized eggs from transgenic worms expressing the indicated fluorescence stress reporter were grown on OP50 or RNAi plates at 20 °C. For N-acetylcysteine (NAC) and mitoquinol (MitoQ) experiments, L4 larvae were transferred on OP50 plates with UV-killed bacteria and supplemented with 10 mM NAC (Sigma-Aldrich, St. Louis, MO, USA) or 7.5 μM MitoQ (Cayman, Ann Arbor, MI, USA), respectively, using the corresponding solvent in control plates. In all cases, microscopic analysis was performed in 1-day adults, considering the slight developmental delay in *lonp-1* mutants. Worms were immobilized with levamisole (Sigma-Aldrich, St. Louis, MO, USA) and mounted on 2% agarose pads on glass microscope slides. Images were captured by fluorescent microscopy using a Leica DMRA upright fluorescent microscope equipped with a Hamamatsu ORCA-flash 4.0 camera and 10× or 40× objectives. All strains were assayed in parallel, and microscopy settings were kept stable throughout each experiment. Fluorescent reporter activity was evaluated through measurement of the average pixel intensity with ImageJ 1.52p (Fiji, https://imagej.net/software/fiji/, accessed on 10 December 2021) in the whole worm image captured. In all cases, approximately 50 worms per strain and condition were used in three biological replicates, and the mean of calculated values was plotted. For nuclear localization of DAF-16, approximately 30 transgenic animals of each genotype expressing *muIs71[daf-16ap::gfp::daf-16a*] were fixed with 4% paraformaldehyde in 1× PBS for 10 min and washed twice with M9 buffer prior to visualization, to avoid DAF-16 translocation as result of handling. Images were captured by confocal microscopy using a Leica TCS SP5 II laser scanning confocal imaging system on a DM6000 CFS upright microscope and a 10× objective. GFP::DAF-16α positive nuclei were counted manually using maximum intensity projections of z-stacks generated in ImageJ 1.52p (Fiji). Shown images are single optical sections.

### 2.7. ROS Measurement

Endogenous hydrogen peroxide levels were measured in transgenic jrIs1[*rpl-17p::HyPer*] worms expressing a HyPer probe in wt or *lonp-1(ko)* mutant background. Approximately 50 worms of the indicated genotypes at day 1 of adulthood were mounted on slides and were used as described in Back et al. [50]. The oxidized and reduced form of HyPer (circularly permutated YFP) of individual worms was excited at 490 and 405 nm, respectively, with a single emission peak at 535 nm. Hydrogen peroxide levels were measured as the ratio of oxidized to reduced HyPer intensity. For Dihydroethidium (DHE) staining, the protocol used was adapted from Aspernig et al. [51]. In brief, approximately 100 1-day adult worms of each genetic background were collected into a 1.5 mL microcentrifuge tube and washed 3 times in M9 buffer. Subsequently, they were stained in 10 μM DHE (Cayman, Ann Arbor, MI, USA, in M9 plus 0.01% PEG) for 2 h on a shaking platform at room temperature. Each tube was wrapped with aluminum foil to prevent the oxidation of dye. After staining, worms were washed in M9 and mounted on 2% agarose pads with levamisole for immediate imaging (the DHE-derived fluorescence intensity is stable over the first 30 min). Unstained worms were imaged to assess the autofluorescence signal, which should be subtracted. Fluorescence density was measured by using ImageJ software.

### 2.8. MitoTracker Staining

For MitoTracker Green staining, synchronized 1-day adult worms were washed into 100 μL M9 supplemented with boiling-killed OP50 bacteria. Subsequently, MitoTracker Green (Invitrogen, Waltham, MA, USA) was added to a final concentration of 10 μM, and worms were incubated at 20 °C overnight (the tube was wrapped with aluminum foil to prevent the oxidation of dye). The next day worms were washed with M9 and placed on NGM with OP50 plates for 1 h to remove excess dye. For Mitotracker Red CMXRos (Invitrogen, Waltham, MA, USA) staining, the same method was performed with 1 h incubation to a final concentration of 4.70 μΜ working solution followed by 2–3 washes with M9 for destaining. Imaging was performed with 40× or 63× magnification in muscle areas using levamisole as an anesthetic agent.

### 2.9. RNA Extraction and Quantitative Reverse Transcription PCR (qRT-PCR)

Total RNA was isolated from frozen worm pellets (200–300 worms per sample) of the indicated genetic background, age, and treatment using TRI Reagent (Sigma-Aldrich, St. Louis, MO, USA). At least three biological replicate samples were harvested and analyzed independently in each experiment. The quality and quantity of RNA samples were determined using Nanodrop 2000 Spectrophotometer (Thermo Scientific, Waltham, MA, USA) before and after DNAse I (Thermo Scientific, Waltham, MA, USA) treatment, according to the manufacturer’s instructions. Reverse transcription using random primers was carried out with FIREScipt RT cDNA Synthesis KIT (Solis BioDyne, Tartu, Esthonia), and quantitative PCR was performed using KAPA SYBR FAST Universal Kit (Kapa Biosystems, Wilmington, MA, USA) in the MJ MiniOpticon system (BioRad, Hercules, CA, USA). Total RNA was extracted from cells with TRIzol™ (Invitrogen, Waltham, MA, USA) according to the manufacturer’s instructions, and first-strand cDNA synthesis was performed with MMLV reverse transcriptase (Invitrogen, Waltham, MA, USA) in 20-μL reactions, using 1000 ng RNA as template and oligo-dT primers. qRT-PCR experiments were performed in 20 μL reactions with the KAPA SYBR FAST qPCR Master Mix 2× kit (Kapa Biosystems, Wilmington, MA, USA) in an MX3000P cycler (Stratagene, La Jolla, CA, USA). Relative amounts of mRNA were determined using the comparative Ct method for quantification, and each sample was independently normalized to its endogenous reference gene (*ama-1* for worms or β-ACTIN for cells). Gene expression data are presented as the mean fold change of all biological replicates relative to control. Primer sequences used for qRT-PCR are shown in Appendix A.

### 2.10. Stress Sensitivity Assays

Heat shock assays were performed by shifting synchronous populations of approximately 100 adult worms from 20 °C to 35 °C for the indicated time points. For RNAi experiments, worms were grown on RNAi plates from eggs before being shifted to 35 °C as 1-day adults. Survival was scored after 16 h of recovery at 20 °C. For the osmotic stress assay, 1-day adults were placed on plates containing 500 mM NaCl for 24 h. For oxidative stress with paraquat (Sigma-Aldrich, St. Louis, MO, USA) or antimycin A (Sigma-Aldrich, St. Louis, MO, USA), appropriately synchronized worms of each genotype were grown in the presence of 40 µM FUdR until the late L4 stage. The next day, 1-day adults were placed on new plates seeded with UV-killed bacteria and containing the oxidant without FUdR. Survival was monitored after 48 h on plates with 30 mM paraquat or after 24 h on plates with 40 μM antimycin A. H_2_O_2_ stress assays were performed in 1-day adults, washed from OP50 plates, into 1.5 mL tubes containing 10 mM H_2_O_2_ (from a stock solution 30% *w*/*w* in H_2_O, Sigma-Aldrich, St. Louis, MO, USA) in a total volume of 1 mL M9. Tubes were incubated for 30 min on a rotating wheel at room temperature. After centrifugation, the worm pellet was transferred to fresh OP50 seeded NGM plates and left for recovery at 20 °C for 30 min before survival was scored. For the tBHP stress tolerance assay, 1- or 2-day adult worms of the indicated genotype were placed on freshly prepared tBHP plates (according to Ewald et al. [52]) containing 7.5 or 10 mM tBHP (from a stock solution of 70% in water, Sigma-Aldrich, St. Louis, MO, USA), without bacteria. Worms were continually repositioned into the center of the plate for the first two hours as they tried to “escape” from the plate and avoid tBHP, and survival was scored every hour. Exploded or damaged animals were censored from the statistics. For sodium arsenite and rotenone stress assays, 1-day adult worms of the indicated genotype were placed on NGM plates seeded with UV-killed bacteria and 7.5 mM sodium arsenite (Acros Organics, Waltham, MA, USA) or 10 µM rotenone (Sigma-Aldrich, St. Louis, MO, USA), and the percentage of worms surviving was determined after 24 h. To assay for sodium azide toxicity, 1-day adults were picked to NGM plates containing 1.5 mM sodium azide (Merck, Darmstadt, Germany), where they remained anesthetized for 24 h. The number of surviving worms was counted after 20 h recovery at 20 °C. For RNAi experiments, synchronized worms grew for one generation on RNAi plates prior to picking them as 1-day adults onto new plates for each assay. All experiments were performed at least three times, and each biological replicate comprised three plates with 30–40 worms per condition and genotype. The percentage survival for each replicate was plotted, and an unpaired *t*-test was used to assess significance.

### 2.11. Protein Extraction and Western Blotting

Approximately 200–300 age-synchronized worms of each strain were washed in M9 buffer. 5X SDS sample buffer (62.5 mM Tris-HCl, pH 6.8, 20% *v*/*v* glycerol, 10% *w*/*v* SDS, 0.1% *w*/*v* bromophenol blue, and 5% *v*/*v* 2-mercaptoethanol) was added to pellets to a total volume of 30 μL and worms were frozen. Worm pellets were boiled for 5 min before loading onto 10% or 12% SDS-PAGE gels. Cells were washed twice with ice-cold 1× PBS and lysed in RIPA buffer (50 mM Tris-HCl, pH 7.4, 1 mM EDTA, 150 mM NaCl, 0.25% *w*/*v* Sodium Deoxycholate, 1% *v*/*v* Triton X-100, 0.1% *w*/*v* SDS and 1 mM PMSF). Protein concentration was measured with BCA Protein Assay Kit (Thermo Scientific, Waltham, MA, USA), and 20 or 30 μg of cell culture protein extracts were separated in 10 or 12% SDS-PAGE gels and subsequently (electro-)transferred onto nitrocellulose blotting membranes (Amersham, GE Healthcare Life Sciences, Chicago, IL, USA). Membranes were blocked in 1XTBS-T containing 5% non-fat milk for 1 h at room temperature. Western blots were performed with the primary antibodies: Anti-LONP1 (Sigma-Aldrich, St. Louis, MO, USA, 1:4000), Anti-ATF4 (Cell Signaling, Danvers, MA, USA, 1:1000), Anti-HSP70 (Cell Signaling, Danvers, MA, USA, 1:1000), Anti-PARP (Cell Signaling, Danvers, MA, USA, 1:500), Anti-ATP6 (Elabscience, Houston, TX, USA, 1:1000), Anti-β-ACTIN (Cell Signaling, Danvers, MA, USA, 1:1500), and the secondary antibodies Anti-Rabbit (Santa Cruz Biotech, Dallas, TX, USA, 1:5000) or Anti-Mouse (Millipore, Burlington, MA, USA, 1:2000). Primary antibodies were added for overnight incubation at 4 °C. Secondary antibodies were incubated for 1 h at room temperature, while the immuno-reacting protein bands were visualized by ECL (Amersham, GE Healthcare Life Sciences, Chicago, IL, USA).

### 2.12. MTT Assays

WM266-4 and HT1080 cancer cells were seeded at a density of 7 × 10^3^ per well into 96-well plates and treated with 1 μM CDDO-Me for 24 h or after LonP1 silencing. Cells were incubated in 3-(4,5-dimethylthiazol-2-yl)-2,5-diphenyltetrazolium bromide (MTT, Sigma-Aldrich, St. Louis, MO, USA) solution. Absorbance was measured at 550 nm, using measurement at 630 nm as a reference, by an Infinite M200 plate reader (Tecan Group Ltd., Männedorf, Switzerland).

### 2.13. Cell Cycle Analysis

CDDO-Me effect on WM266-4 and HT1080 cell cycle progression was assessed via the FACS approach. Briefly, cancer cells were plated in a 60 mm culture dish at a density of 8 × 10^5^ cells per dish. The next day, 1 μM CDDO-Me treatment was applied for 24 h. Adherent cells were collected with trypsinization, fixed in 70% ethanol and stained with propidium iodide (PI) solution (50 μg/mL) containing 250 μg of DNAse-free RNAse A. Finally, they were analyzed using flow cytometry in a Beckton Dickinson’s FACScalibur (Franklin Lakes, NJ, USA) at 542 nm, and the Modfit program.

### 2.14. Scratch-Wound Assays

WM266-4 and HT1080 cancer cells were plated in a 60 mm culture dish (Greiner, Kremsmünster, Austria) at a density of 8 × 10^5^ cells per dish. When cells reached 80–90% confluency, cell monolayers were scratched using a 200 μL-tip. Cells were rinsed with warm 1× PBS and then incubated with 1× DMEM (Dulbecco’s modified Eagle’s medium, Biowest, Nuaillé, France) supplemented with 2% FBS (fetal bovine serum, Biowest, Nuaillé, France) and 1% Penicillin/Streptomycin (Gibco) with 500 nM CDDO-Me (Cayman Chemicals, Ann Arbor, MI, USA). Cell images were taken under an inverted Nikon Eclipse microscope Ts2 equipped with a Basler Microscopy ace 2.3 MP camera at 0 and 24 h after incubation, using the 4× objective lens. Images were acquired using the Basler Microscopy Software 2.1 (Build 17017).

### 2.15. Statistics

Graphs and statistical analysis were performed with GraphPad Prism version 8.0.0 for Windows (GraphPad Software, San Diego, CA, USA, www.graphpad.com/scientificsoftware/prism, accessed on 10 December 2021). Statistical analysis was performed by comparing each sample to the appropriate control in the same condition and *p* values were determined by Student’s *t*-test (unpaired or paired as indicated) and depicted as follows: **** *p* < 0.0001; *** *p* = 0.0001–0.001; ** *p* = 0.001–0.01; * *p* = 0.01–0.05; ns indicates not significant with *p* value ≥ 0.05. For analyses involving multiple strains and conditions, a two-way ANOVA with Tukey’s multiple comparisons analysis was used to assess the significance and differences between groups.

## 3. Results

### 3.1. Mitochondrial Lon Protease Supports Normal Development, Fecundity, and Lifespan of C. elegans

The worm homolog of human LonP1 is encoded by the *lonp-1* (C34B2.6) gene, which lies within an operon containing four genes in a row, namely *sdha-2*(C34B2.7), *spcs-1*(C34B2.10), *lonp-1*(C34B2.6) and *ttc-1*(C34B2.5) (Figure 1A, https://www.wormbase.org, last accessed on 10 December 2021). The *C. elegans* LONP-1 protein consists of 971 amino acids and shares 47.6% identity and 62.1% similarity with the human protein (Appendix A). The worm homolog has a typical Lon protease structure with an N-terminal mitochondrial targeting sequence (MTS), the central AAA^+^ domain, and the C-terminal proteolytic domain bearing the Serine-Lysine catalytic dyad (Figure 1B; [1]). While the central core domain with the ATPase motif and the sensor- and substrate-discrimination (SSD) domain is highly diverse among the Lon homologs from distinct species, the human (HsLonP1), bacterial (EcLon), Arabidopsis (AtLon1), and worm (CeLonP1) proteases showed similar structure, suggesting conserved functions (Figure 1B; [2]). An uncharacterized *lonp-1(tm5171)* mutant, with a 5 bp insertion and a 490 bp deletion encompassing the third exon and part of the fourth exon of the *lonp-1* gene (Figure 1B), was found to generate a low-abundance truncated transcript that could lead to a mitochondrial peptide carrying only the first 108 residues of the full-length LONP-1 protein (Appendix A). While *tm5171* is a *lonp-1* null allele, to avoid possible secondary effects on mitochondrial homeostasis, CRISPR/Cas9 was applied to generate *lonp-1(ko)* for further analysis. The absence of LONP-1 protein in this strain was verified by Western blot using antibodies against human LonP1 (Figure 1A).

Although in *Drosophila* and mouse, disruption of LonP1 results in embryonic lethality [14,53], both *C. elegans lonp-1(ko)* and *lonp-1(tm5171)* mutants were viable but displayed poor synchrony in development and a delayed pace of larval development, compared with wild-type (wt) animals (Figure 1C and Appendix A). Furthermore, the mutants showed reduced total brood size, with a slightly extended egg-laying period (Figure 1D and Appendix A). Mitochondrial mutants with defects in respiration and metabolic activity often exhibit low rates of growth and proliferation, while they are frequently associated with lifespan extension. However, both *lonp-1* mutants had significantly shorter mean lifespan than wt worms, at 20 °C (Figure 1E) and 25 °C, on OP50 *E. coli* diet, regardless of the presence of 5′-fluorodeoxyuridine (FUdR) that is commonly used to prevent egg hatching or progeny growth (Appendix A). These phenotypes were attributed to *lonp-1* deletion, given that expression of a *lonp-1::gfp* fusion transgene, driven by the internal promoter of *lonp-1* within the operon, could partially rescue the impaired growth rate and short lifespan of *lonp-1* mutants (Appendix A). The expression of the *lonp-1::gfp* transgene was low but ubiquitous, and the subcellular localization of the produced LONP-1::GFP protein was typical of mitochondrial proteins, as confirmed by staining of transgenic worms with vital dye MitoTracker Red CMXRos (Appendix A).

### 3.2. Mitochondrial and ROS Homeostasis Are Distorted in Lonp-1 Mutants

Mammalian cells lacking mitochondrial LonP1 protease exhibit impaired mitochondrial respiration and reduced membrane potential [7,54]. To evaluate the effects of *lonp-1* disruption on the mitochondrial network, worms were stained with MitoTracker Green, a vital dye that labels all mitochondria, versus MitoTracker Red CMXRos, which depends on mitochondrial membrane dynamics and binds to internal mitochondrial components. In *lonp-1(ko)* mutants MitoTracker Green revealed the disturbed morphology of mitochondria that were slightly swollen, disorganized, and fragmented. Likewise, staining of mitochondria with MitoTracker Red CMXRos was significantly decreased compared with wt animals (Figure 2A), suggesting impairment of membrane potential in *lonp-1* mitochondria. In addition, *lonp-1* mutants had reduced mitochondrial mass as assessed by the *myo-3_p_::GFP^mt^* reporter (Figure 2C). While in wt animals this reporter detected the normal network of mitochondria, usually aligned with the myofibrils [55] and colocalized with MitoTracker Red CMXRos, this pattern was distorted in *lonp-1* young adults (Figure 2B). Consistent with the disruption of mitochondrial dynamics known to affect animal behavior [56,57], the *lonp-1* worms showed considerably reduced sinusoidal body bends and thrashing rates, together with bacterial-aversive behavior (Appendix A). Mitochondrial dysfunction in *lonp-1(ko)* mutants was further confirmed (Figure 2D) by using a xenobiotic *cyp-14A4p::gfp* reporter that portrays the induction of a *C. elegans* cytochrome P450 gene in response to mitochondrial damage [58].

Impaired folding or degradation of matrix proteins in *lonp-1* mutants could be associated with increased ROS production, which would further exacerbate mitochondrial morphology and function [59]. In support of this hypothesis, staining of worms with the ROS-sensitive dye dihydroethidium (DHE) [60,61] demonstrated increased cytosolic superoxide levels in *lonp-1* versus wt animals (Figure 2E). In addition, to monitor hydrogen peroxide (H_2_O_2_) levels in vivo, the fluorescent H_2_O_2_ redox sensor HyPer, under the control of a ubiquitous promoter, was used [50]. In fact, an increased ratio of oxidized to reduced HyPer in *lonp-1* compared with wt adults was assessed, indicating higher endogenous peroxide levels in mutant animals (Figure 2F).

### 3.3. Activation of ATFS-1-Mediated Retrograde Response in Lonp-1 Mutants

Increases in endogenous levels of ROS have been shown to evoke the UPR^mt^ signaling through the activity of the ATFS-1 transcription factor [62]. In addition to this, stress conditions that alter mitochondrial function promote cytosolic accumulation and nuclear import of ATFS-1, whereas, under unstressed conditions, mitochondrial ATFS-1 turnover has been linked to LONP-1 protease activity [32]. However, RNAi-mediated knockdown of *lonp-1* in wt animals was not able to activate UPR^mt^ or impair worm development [32]. Using two well-established UPR^mt^ markers, the nuclear-encoded mitochondrial chaperone *hsp-60*/mtHSP60 and the *hsp-6*/mtHSP70 [26], we evidenced that both were constitutively upregulated in *lonp-1* mutants. Expression levels of the endogenous *hsp-6* and *hsp-60* genes, together with the fluorescence levels of the corresponding *hsp-6p::gfp* and *hsp-60p::gfp* reporters, were significantly induced in *lonp-1* compared with wt animals (Figure 3A,B). The discrepancy in the induction of the UPR^mt^ markers or worm development between *lonp-1* deletion and *lonp-1(RNAi)* possibly arises from the inefficient RNAi-mediated silencing of the *lonp-1* gene, which is not adequate to induce UPR^mt^. We confirmed this using a *lonp-1(RNAi)* clone in wt worms carrying the *hsp-6p::gfp* reporter (data not shown).

Induction of the fluorescence levels of *hsp-6p::gfp* in *lonp-1* mutant strains was dependent on ATFS-1 transcriptional activity, as feeding worms with *atfs-1(RNAi)* suppressed this induction (Figure 3C). However, the homeodomain-containing transcription factor DVE-1 and the small ubiquitin-like protein UBL-5, both of which are known to work in parallel to *atfs-1* in the nucleus to activate UPR^mt^ [24], did not disrupt the induction of *hsp-6p::gfp* in *lonp-1* worms (Figure 3C). The mitochondrial matrix protease CLPP-1/ClpP and the mitochondrial inner-membrane localized peptide transporter HAF-1 have also been linked to UPR^mt^ signaling [24]. Nevertheless, the *lonp-1* deficiency was sufficient to induce the activation of *hsp-6p::gfp* by a *lonp-1* deletion in a *clpp-1-* and *haf-1*-independent manner (Appendix A). The integrated stress response pathway (ISR) is required for UPR^mt^ induction upon mitochondrial dysfunction in mammals but is dispensable for induction of the UPR^mt^ in worms [42]. The *atf-4* (T04C10.4) gene (previously named *atf-5*, [63]) encodes for the single worm homolog of ATF4 and ATF5 bZIP transcription factors that are induced when translation is suppressed in mammalian ISR. Using an established ISR marker that expresses the intact *atf-4* gene, including the upstream ORFs, fused with *gfp* [63], we showed the lack of reporter’s induction in *lonp-1(ko)* worms (Appendix A), excluding global translation suppression in these mutants.

To evaluate the impact of *atfs-1* deletion on *lonp-1* physiology and lifespan, we made many attempts to combine the loss-of-function *atfs-1(gk3094)* allele with each *lonp-1* mutant allele. Although the two genes are located in different chromosomes, we were unable to obtain fertile homozygous *atfs-1;lonp-1* offspring in crosses and only sterile double mutant animals were generated. Likewise, sustained (for two or more generations) RNAi-mediated knocking down of *atfs-1* in *lonp-1* mutants resulted in embryonic lethality and few progenies with delayed larval development, in contrast to wt worms where *atfs-1(RNAi)* had no discernible effect (Appendix A). Importantly, mutants of *lonp-1* displayed elevated mRNA levels of *atfs-1* (Appendix A), perhaps due to positive self-regulation of *atfs-1* expression by ATFS-1 itself [33]. RNAi of *atfs-1* post-developmentally (at L4 stage or young adult stage) shortened the lifespan of *lonp-1* but did not impact the lifespan of wt (Appendix A). These data support an essential role of *atfs-1* in the development and lifespan determination of *lonp-1* mutants, with ATFS-1 activity being important for worms to encounter mitochondrial proteotoxic stress induced by loss of *lonp-1* protease.

### 3.4. LONP-1 Deficiency Induces Cytosolic Oxidative Stress Responses

Increased levels of ROS have also been demonstrated to elicit activation of cytosolic stress signaling pathways and transcription factors that regulate the expression of ROS-detoxifying and protective systems. In *C. elegans*, SKN-1, the worm ortholog of the mammalian NRF1/2 transcription factor, coordinates a Phase II-like cellular antioxidant response [64,65,66,67] and is a transcriptional target of ATFS-1 under mitochondrial stress [33]. In agreement with this, 1-day adult *lonp-1* mutants exhibited elevated mRNA levels of *skn-1* and of two SKN-1 target genes, the glutathione S-transferases *gst-4* and *gst-13* [68,69] (Figure 4A). Using a *gst-4p::gfp* reporter [70], we confirmed the induction of *gst-4* in *lonp-1* deficient worms and treatment with hydrogen peroxide (H_2_O_2_), a known ROS-generator substantially enhanced fluorescence in mutant compared with wt animals (Figure 4B). Depletion of *skn-1* diminished the *gst-4p::gfp* signal in both *lonp-1* and wt backgrounds; depletion of *atfs-1* mitigated the signal only in *lonp-1* mutants (Figure 4C), whereas RNAi against *dve-1* or *ubl-1* UPR^mt^ factors rather increased the levels of the *gst-4::gfp* reporter (Appendix A). Furthermore, treatment of worms with the antioxidant N-acetylcysteine (NAC), which quenches all types of ROS [71], interfered with the induction of *gst-4p::gfp* to a greater extend in *lonp-1* compared with its effect in wt animals (Appendix A). Likewise, the application of a mitochondrial-targeted antioxidant mitoquinol (MitoQ) completely abolished the induction of reporter in *lonp-1* mutants, reaching the levels of untreated wt worms (Appendix A). Taken together, our data point to an interaction of ATFS-1 activation and ROS production in *lonp-1* mutants for upregulation of *skn-1* and its target genes. In stark contrast, RNAi against *skn-1* and NAC or MitoQ supplement had no effect on the fluorescence intensity of *hsp-6p::gfp* reporter, indicating that ATFS-1 induces *hsp-6* independently of ROS levels (Appendix A).

In several mitochondrial mutants, ROS accumulation can activate the forkhead transcription factor DAF-16/FOXO of the Insulin/IGF signaling pathway, and ATFS-1 has been shown to affect the expression of DAF-16 target genes, at least partially through the nuclear translocation of DAF-16 [28,72]. To monitor in vivo the localization of DAF-16 in *lonp-1(ko)* mutants, transgenic worms expressing the reporter *muIs71[daf-16ap::gfp::daf-16a*] were examined under normal growth conditions and in response to mild heat stress, which triggers nuclear translocation of DAF-16. In all conditions, loss of *lonp-1* increased GFP::DAF-16α nuclear accumulation (Figure 4D), and this was evident even though endogenous DAF-16 protein was present, as we were unable to generate a *lonp-1;daf-16* deletion strain due to the close linkage (0.02 mu) of these two genes on chromosome I. Mitochondrial MnSOD *sod-3*, metalloproteinase *mtl-1,* and small heat-shock protein *hsp-16.2* are *bona fide* DAF-16 target genes [73,74,75]. Consistent with the enhanced nuclear import of DAF-16, all three genes were upregulated in *lonp-1* mutants compared with wt animals (Figure 4D). Accordingly, in worms expressing a *sod-3p::gfp* transgene, increased fluorescence was observed upon loss of *lonp-1* (Appendix A). This induction was attenuated upon *atfs-1(RNAi),* but decreasing ROS with NAC did not compromise activation of the reporter (Appendix A), supporting a role of ATFS-1 in DAF-16 activation in *lonp-1* mutants.

### 3.5. Induction of the Heat Shock Response (HSR) in Lonp-1 Mutants

The heat shock response (HSR) is an adaptive mechanism in eukaryotes, activated by various stressors, such as high temperature, oxidative stress, and heavy metals, and protect cells from protein denaturation, misfolding, and aggregation. Heat shock transcription factor 1 (HSF1) induces the expression of a group of molecular chaperones, known as heat-shock proteins (*hsp*), which assist in cytosolic protein folding during stress. In the *lonp-1* mutant background, we measured transcriptional upregulation of the small *hsp-16.2* and *hsp-16.1*, as well as the two HSP70 family members *hsp-70*(C12C8.1), and *hsp-70*(F44E5.4), under normal growth temperature (Figure 5A). On the contrary, there was no transcriptional induction of *hsf-1*, *hsp-1*/HSC70, *daf-21*/HSP90, or the *hsp-3* and *hsp-4* chaperones, the two homologs of the mammalian HSP70 member BiP that are associated with endoplasmic reticulum UPR (UPR^ER^) (Figure 5A). Using an *hsp-16.2p::gfp* reporter as a readout of cytosolic HSR, an enhanced signal in the *lonp-1* worm, compared with wt, was observed under mild heat stress where fluorescence of the reporter is measurable (Figure 5B). As showed by NAC supplementation, this effect is mediated, at least in part, by excessive ROS production in *lonp-1* mutants and requires heat-shock factor 1 (HSF-1) activity but not DAF-16 or ATFS-1 (Appendix A). However, it was not possible to assess the requirement of *atfs-1* in the absence of heat stress, as *atfs-1(RNAi)*-treated wild-type worms exhibit enhanced expression of both *hsp-16.2p::gfp* reporter and endogenous *hsp-16.2* gene [28,76,77], most likely caused by enhanced mitochondrial stress. Nevertheless, activation of the HSR under unstressed conditions indicates an accumulation of misfolded and damaged proteins in the cytoplasm of *lonp-1* deficient cells.

### 3.6. Specific Responses of Lonp-1 Mutants to Exogenous Stresses

LonP1 is a crucial stress-responsive protein, vital for cell survival [78], but a *lonp-1* loss in worms is able to induce the activity of key transcription factors and genes involved in stress defense and adaptation. Therefore, we explored the effects of *lonp-1* absence in counteracting various forms of stress. Following exposure to acute heat shock (HS), *lonp-1* mutants were far more resistant to heat stress compared with wt controls at day 1 of adulthood (Figure 5C). Because heat stress response in *C. elegans* is attenuated at the onset of egg laying [79], the enhanced thermotolerance of *lonp-1* mutants was further confirmed at day 3 of adulthood (Figure 5C). Additionally, *lonp-1* animals displayed extreme resistance to the high osmolarity of NaCl (Figure 5D) and increased tolerance to toxicity induced by the pro-oxidant paraquat or antimycin A, which induce superoxide anion generation in mitochondria [26,57,62,80] (Figure 5E). Also, *lonp-1* adults exhibited increased survival compared with their wt counterparts after acute treatment with H_2_O_2_ or an organic peroxide named tert-butyl hydroperoxide (tBHP) (Figure 5F). Intriguingly, *lonp-1* mutants were sensitive to three other inhibitors that can induce ROS levels by several mechanisms, sodium arsenite [81], rotenone [82], and sodium azide [83] (Figure 5G), suggesting that additional stresses or metabolic changes [84] take place following exposure to these chemicals, which *lonp-1* mutants cannot overcome. The expression of the *lonp-1::gfp* transgene in *lonp-1* mutants was sufficient to alleviate the sensitivity of mutants in all these oxidants (Appendix A). Thus, our data unveil specific *lonp-1*-environment interactions, with a *lonp-1* deficiency affecting condition-specific defenses during the organismal oxidative stress response. Indeed, RNAi-mediated knockdown of *skn-1* or *atfs-1* significantly compromised the tolerance of wt and *lonp-1* adults to tBHP, but only *atfs-1(RNAi)* diminished the resistance of *lonp-1* to antimycin A (Appendix A). Finally, the extreme heat and hypertonic stress resistance were found to be independent of *atfs-1,* and *skn-1* transcription factors and NAC treatment did not compromise thermotolerance of *lonp-1* mutants (Appendix A).

### 3.7. Treatment of Worms with CDDO-Me Induced Lonp-1-Specific Responses

CDDO-Me, the methyl ester derivative of CDDO triterpenoid, has been shown to bind and selectively inhibit the protease activity of LON in human cells [85], inducing mitochondrial protein aggregation, fragmentation of mitochondria, and apoptosis in colon cancer cells [86]. We first treated *C. elegans* with this triterpenoid and observed dose-dependent retardation in the growth of both wt and *lonp-1* mutants, indicating that both can take up the compound from the plates’ surface (Appendix A). Next, we analyzed the expression of the stress-responsive genes in wt worms treated with CDDO-Me either from eggs (long-term) or at the end of the L4 stage for 24 h (short-term). In both conditions, there was a robust induction of cytosolic small *hsp* and *hsp-70* genes, but not of ER-specific *hsp-3* chaperone, compared with DMSO-treated controls (Figure 6A). Long-term treatment of wt worms with CDDO-Me led to further up-regulation of antioxidant genes, such as *gst-4* and *gst-13* (*skn-1*-target) or *mtl-1* (*daf-16*-target), without affecting the expression levels of *skn-1* or *atfs-1* (Figure 6B). Interestingly, this up-regulation of antioxidant genes was not observed in the *lonp-1* mutant background, where a significantly smaller induction of *hsp* genes occurred (Figure 6C). Although the cellular mechanism of action and the specificity of CDDO-Me are ambiguous, our data support a hypothesis in which these responses could have been triggered by a possible inhibitory function of CDDO-Me on LONP-1.

Compatible with the induced HSR genes, CDDO-Me profoundly improved the tolerance of 1-day wt adults to heat stress and did not further enhance the heat resistance of *lonp-1* mutants (Appendix A). The enhanced thermotolerance of CDDO-Me-treated worms was *atfs-1*-independent, as handling a loss-of-function *atfs-1(gk3094)* mutant strain with CDDO-Me resulted in HS resistance as well (Appendix A). In addition, treatment with CDDO-Me increased survival of wt but not of *lonp-1* mutants to oxidative stress induced by tBHP (Appendix A). Intriguingly, exposure of wt and *lonp-1* worms to CDDO-Me was not able to induce the UPR^mt^, as measured by the mRNA levels of *hsp-6* and *hsp-60* chaperones (Figure 6D). In support of this, treatment of wt worms with CDDO-Me did not induce alterations in mitochondria morphology and dynamics, as assessed by MitoTracker Red staining (Appendix A). Combined, these data indicate that CDDO-Me, contrary to *lonp-1* gene deletion, is not able to activate UPR^mt^ in worms. Even though treatment with CDDO-Me did not alter the endogenous levels of LONP-1 protein (Appendix A), the fact that the transcriptional stress responses are not evident when LONP-1 is missing indicates a possible role of CDDO-Me in the induction of mito-cytosolic proteostasis mechanisms in cross talk between LONP-1 activity in the mitochondrial matrix and the cytoplasm.

### 3.8. Effects of LONP1-Deficiency and CDDO-Me Treatment in Human Cancer Cells

To corroborate our results from gene expression analysis in *lonp-1* mutant worms, we surveyed a number of changes in response to mitochondrial LonP1 downregulation in human cancer cells. Due to their altered proliferation rate, metabolism levels, signaling pathways activities, etc., cancer cells are under permanent stress. In order to survive, they need to somehow balance the constant presence of stress through the activation of a variety of response mechanisms. These include the mitochondrial and endoplasmic reticulum unfolded protein responses (UPR^mt^, UPR^ER^), the integrated stress response (ISR), the heat shock response (HSR), the antioxidant response (AR), etc. To study the effect of LonP1 disruption in cancer cells, we tested the types of responses generated by LonP1 downregulation, first by genetic knockdown and then using CDDO-Me to pharmacologically inhibit the protease. To this end, we have used two cell lines belonging to different types of cancer bearing-unlike characteristics, but both overexpressing LonP1 (Appendix A). HT1080 fibrosarcoma cells are known to be homozygous for CDKN2A deletion, to have an activated N-ras oncoprotein (p.Gln61Lys), and carry a p.Arg132Cys IDH1 mutation, whereas WM266-4 metastatic melanoma bear a PTEN hemizygous deletion and a BRAF p.Val600Asp mutation (https://web.expasy.org/cellosaurus/, last accessed on 1 December 2021).

Genetic knockdown of LonP1 (Figure 7A) was found to elicit transcriptional changes in a number of genes belonging to ISR, UPR^mt^, HSR, and antioxidant response in HT1080 cells (Figure 7B). Intriguingly, WM266-4 melanoma cells exhibited a different general pattern of expression, displaying activation of ATF5 but no activation of the other pathways (Figure 7C). On the other hand, pharmacological inhibition of LonP1 with the use of CDDO-Me caused an evident transcriptional activation of all stress response mechanisms studied here in both cancer cell lines (Figure 8A,B). Similarly, upregulation of the ISR-related transcription factor ATF4 and the HSR-specific cytoplasmic chaperone HSP70 was confirmed in both cell lines after CDDO-Me treatment (Figure 8C), whereas, after LonP1 siRNA, only ATF4 in HT1080 cells was found to be upregulated at the protein level (Appendix A). Nevertheless, taken together, these findings suggest the existence of tight communication events between mitochondrial and cytoplasmic stress response in cancer cells, caused by CDDO-Me-induced LonP1 inhibition, but also after LonP1 disruption by siRNA-induced knockdown, although here the response was less prominent.

On top of the above, siRNA-mediated as well as pharmacological inhibition of LonP1 were found to interfere with the metabolic activity rate of mitochondria of both lines, as shown by the decreased relative percentages in the MTT assays, known to be based on measurements of the mitochondrial metabolic activity of cells (Figure 9A). Moreover, Western blots performed against ATP6 have shown that this alpha subunit of the ATP synthase complex, known to be essential for normal mitochondrial function, and found to be mutant or downregulated in a variety of human diseases of mitochondrial origin, was significantly downregulated in CDDO-Me- and siLonP1-treated cells of both lines (Figure 9B). The above results reflect the presence of mitochondrial dysfunction in both cell lines after LonP1 disruption. In addition, CDDO-Me treatment (at 1 uM) was shown to cause mild apoptotic cell death as evidenced by PARP (poly adenosine diphosphate-ribose polymerase) cleavage in both cell lines, whereas, after LonP1 siRNA, PARP cleavage could be detected only in HT1080 fibrosarcoma cells (Appendix A). Nevertheless, PI/FACS experiments suggested that the levels of apoptosis of CDDO-Me-treated cells were only slightly higher than the controls (Appendix A). Finally, we questioned the effect of LonP1 disruption on cancer cell motility. We found that although LonP1 siRNA was not able to interfere with cell movement, pharmacological inhibition of LonP1 was able to negatively affect the motility of both fibrosarcoma and melanoma cells (Figure 9C).

Overall, genetic, and pharmacological deactivation of LonP1 was demonstrated to impede physiological processes in both experimental systems (*C. elegans* and cancer cells) and induced similar stress-responsive transcriptional changes to adapt mitochondrial dysfunction to specific cell stresses or metabolic demands.

## 4. Discussion

Protein quality control is fundamental for the development and survival of all organisms, and proteomic integrity is preserved by a number of protein defense mechanisms and surveillance pathways that are typically intertwined. Mitochondria are a central hub of cellular metabolism and play crucial roles in integrating signals regulating cell survival. Maintenance of mitochondrial homeostasis involves several mechanisms, such as mitochondria-localized chaperones and proteases, organelle degradation, inter-organellar communication, or intra- and trans-cellular signaling [87]. Although there is significant knowledge on the regulation and function of each mechanism or pathway, mostly at the cellular level, little is known about how they are coordinated at the organismal level and connected to organismal health and aging. Moreover, mitochondrial dysfunction has been implicated in multiple human diseases [88,89].

The ATP-dependent protease LonP1 of the mitochondrial matrix is a highly conserved Serine–Lysine peptidase that plays a major role in mitochondrial protein quality control [90]. It can also regulate mitochondrial RNA metabolism and translation [44,53,91] while supporting cell viability under stress [3]. LonP1 is considered to perform various functions as a chaperone, a DNA-binding protein, and a peptidase with both nonspecific and specific proteolytic activity, even though its overall substrate repertoire remains unclear [92]. Moreover, little is yet known about how LonP1 deficiency affects cellular and organismal physiology or responses to a broad range of challenges. Such aspects still remain obscure, mainly because of the indispensable role of LonP1 in cell survival, as its deletion in *Drosophila* and the mouse causes embryonic lethality [14,53]. The fact that constitutive knockout of other mitochondrial proteases in mice results in normal embryo development and viability emphasizes the crucial role of LonP1 [14] and justifies the growing interest in elucidating its involvement in the above mechanisms. Using two experimental systems, the nematode *C. elegans* and human cancer cells, we analyzed the cellular and organismal transcriptomic stress responses to Lonp1 deficiency caused by genetic and pharmacological approaches and how these affect physiological processes and phenotypes.

Consistent with the well-described role of Lonp1 protease in mitochondrial quality control surveillance, deletion of LONP-1 protease in worms caused mitochondrial structure damage and dysfunction, accompanied by increased levels of ROS. As a result, *lonp-1* null mutants displayed a slower rate of development, smaller brood size, altered behavior, and shorter lifespan than wt animals. Additionally, loss of *lonp-1* was sufficient to induce a retrograde stress response known as UPR^mt^, bypassing the need for CLPP-1 and HAF-1 activities, but was insufficient to induce ISR in worms. Partial inactivation of LonP1 in *Drosophila* also led to behavioral deficits, shortened lifespan, and induction of UPR^mt^ [15,53]. It is known that induction of UPR^mt^ leads to epigenetic modifications and transcriptional changes in a plethora of genes in order to restore mitochondrial function [33]. The activity of the ATFS-1 transcription factor, a critical activator in UPR^mt^, is not essential for worms’ normal growth and lifespan, but it was found to be essential for the viability of *lonp-1* null mutants. Post-developmental downregulation of *atfs-1* could also reduce the lifespan of *lonp-1* animals, signifying that the induction of UPR^mt^ contributes to the lifespan regulation of these mutants.

Besides the induction of UPR^mt^, deletion of *lonp-1* in *C. elegans* initiated a range of cytoplasmic stress-response mechanisms to help worms cope with dysfunctional mitochondria. The activity of ATFS-1 was required for induction of UPR^mt^ in *lonp-1* mutants, independently of ROS accumulation, whereas both ATFS-1 and elevated ROS contributed to SKN-1-dependent oxidative stress response. ATFS-1 was also found to be responsible for nuclear translocation of DAF-16/FOXO transcription factor and the induction of *sod-3p::gfp* reporter in *lonp-1* adults. Although activation of DAF-16 by ROS has been observed in several long-lived mitochondrial mutants and contributes to lifespan extension [72,93], NAC treatment of *lonp-1* mutants rather increased than suppressed the induction of *sod-3p::gfp* reporter, suggesting different mechanisms of DAF-16 activation. Moreover, loss of *lonp-1* significantly enhanced basal expression of HSR genes, such as members of the HSP70 family and small HSPs. This upregulation of HSP genes and antioxidant genes was also observed upon treatment of wt worms with CDDO-Me, a triterpenoid that inhibits, among others, LonP1 in human cells. These results suggest a cross talk between the mitochondrial and cytosolic UPR mechanisms to protect cells from proteotoxic stress. Indeed, deletion of *atfs-1* was previously reported to activate the cytosolic UPR, possibly as a compensatory response for the disruption of UPR^mt^ [28,93].

Enhanced HSR, induced by disruption of several mitochondrial genes, was also shown to prevent the age-dependent collapse of cytosolic proteostasis [94]. Consistent with this, deletion of *lonp-1* significantly increased resistance to heat and other forms of stress, such as high osmolarity induced by high NaCl concentrations or acute oxidative stress induced by paraquat, antimycin, H_2_O_2_, or tBHP. However, *lonp-1* adults exposed to rotenone, sodium arsenite, or sodium azide were more sensitive than wt adults, indicating condition-specific defenses in organismal stress response. It has been reported that pharmacological treatment of worms with various drugs, including paraquat, antimycin A, rotenone, and sodium azide, strongly induce UPR^mt^ [95,96], whereas arsenite activates the mitochondrial antioxidant response but not the UPR^mt^ [96]. Our data demonstrate that the mechanisms which underpin stress resistance in *lonp-1* mutants can be distinct, inducing specific transcriptional programs and metabolic changes. Moreover, even though the ability to cope with stress can have a positive impact on lifespan regulation, these can be uncoupled in *lonp-1* mutants, in agreement with previous studies on other mitochondrial mutants [28,31,97].

In human cancer cells, genetic downregulation of LonP1 by siRNA in HT1080 fibrosarcoma cells was found to cause a variety of transcriptional changes in genes belonging to a wide spectrum of cellular stress–response mechanisms, such as the mitochondrial unfolded protein response, but also the cytoplasm-based pathways of the integrated stress response, heat shock response, and antioxidant response. This suggests that complementary mechanisms are activated upon LonP1 disruption to combat mitochondrial stress. Indeed, LonP1 downregulation in HeLa cells was found to induce effects reminiscent of oxidative stress-related mitochondrial dysfunction, mainly through impairment of energy metabolic functions, such as the glycolytic and the respiratory capacity of mitochondria, and also to cause enhanced NRF2 expression and decreased cell proliferation [98]. However, siRNA-induced LonP1 disruption did not induce any of the above stress response pathways in WM266-4 metastatic melanoma cells, advocating in favor of the presence of additional compensatory mechanisms, owing to specific cellular demands. Similarly, a stable genetic LONP1 knockdown (gKD) in HeLa cells, with a significant reduction in cellular protein levels, did not cause severe alterations in mitochondrial structure and function or strong defects in mitochondrial protein degradation efficiency, at least under normal conditions [99].

On the other hand, pharmacological inhibition of LonP1 using CDDO-Me was found to transcriptionally upregulate almost all genes involved in the mitochondrial and the cytoplasmic stress response pathways described above in both cancer cell lines. The differential effect of CDDO-Me, compared with siLonP1, particularly on WM266-4 cells, indicates the existence of possible additional cellular functions targeted by CDDO-Me [100] or, alternatively, LonP1-independent regulation of mitochondrial function by CDDO-Me [101]. Furthermore, disruption of LonP1 by genetic or pharmacological knockdown caused clear mitochondrial dysfunction in both cell lines, activating an apoptotic cell suicide program. Finally, while LonP1 siRNA was not possible to significantly interfere with cell movement, treatment with CDDO-Me was found to decrease the motility of fibrosarcoma and metastatic melanoma cells, suggesting that LonP1 could be a potential target for cancer therapy in these cells. LonP1 is often overexpressed in aggressive tumors, and LonP1 deficiency has been associated with protection against other cancer types [102].

To conclude, this work highlights the crucial role of LonP1 in cellular and organismal stress response and how disruption of LonP1 activity could influence multiple signaling pathways and factors, to support the adaptation of mitochondrial function to diverse insults. Genetic and pharmacological inhibition of LonP1 has led to experimental system-specific phenotypes but also activation of similar stress responses through retrograde signaling triggered in mitochondria, which diversely modulate organismal stress response, aging, and apoptosis.

## Figures and Tables

**Figure 1 cells-11-01363-f001:**
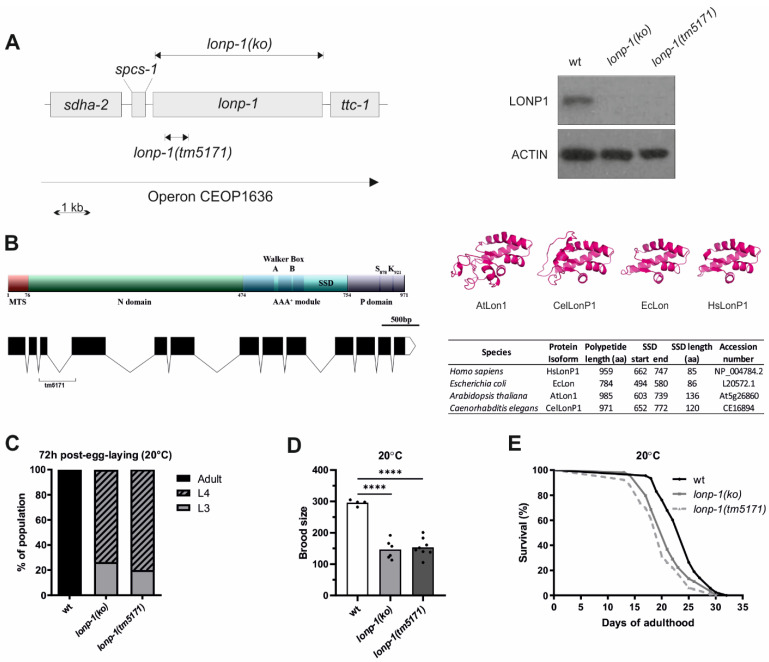
Disruption of *C. elegans* LONP-1 activity impairs normal development, fecundity, and lifespan. (**A**) Schematic of the operon CEOP1636, the CRISPR/Cas9-generating *lonp-1(ko)* null allele, and the *lonp-1(tm5171)* loss-of-function allele. Western blot analysis, using an antibody against human LonP1 and anti-β-ACTIN as a loading control, showed the absence of LONP-1 protein in both *lonp-1* mutant strains. (**B**) Domain structure and functional regions of LONP-1 protein. The mitochondrial target sequence (MTS) directs the translocation of the precursor protein across the mitochondrial membranes; the N domain is involved in substrate recognition, along with the central AAA^+^ module, which contains the Walker Box A and B motifs for ATP-hydrolysis and the sensor- and substrate-discrimination (SSD) domain. The C-terminal proteolytic domain (P domain) contains the serine (S) and lysine (K) catalytic dyad residues. The graphic showing the gene structure of LONP-1 was created using the Exon-Intron Graphic Maker (http://wormweb.org/exonintron, last accessed on 10 December 2021). Black boxes represent exons linked by lines corresponding to introns. The bracket points to the sequences deleted in the *tm5171* allele. In the right panel, the molecular modeling of the SSD domain of human (HsLonP1), bacterial (EcLon), Arabidopsis (AtLon1), and worm (CelLonP1) proteases is presented. (**C**) The growth rate of *lonp-1* mutants 72 h post-egg-laying at 20 °C versus the wild-type N2 animals (wt). At this time point, wt worms have reached the adult stage, while *lonp-1* mutants were at the larval stage 3 or 4 (L3, L4). (**D**) Fertility assays of *lonp-1* mutants showing the mean number of viable progenies per individual in all biological replicates. An unpaired *t*-test was used to assess significance (**** *p* value <0.0001). (**E**) Lifespan assays of *lonp-1* mutants at 20 °C. Replicates and statistical analysis of lifespan assays are shown in Appendix A.

**Figure 2 cells-11-01363-f002:**
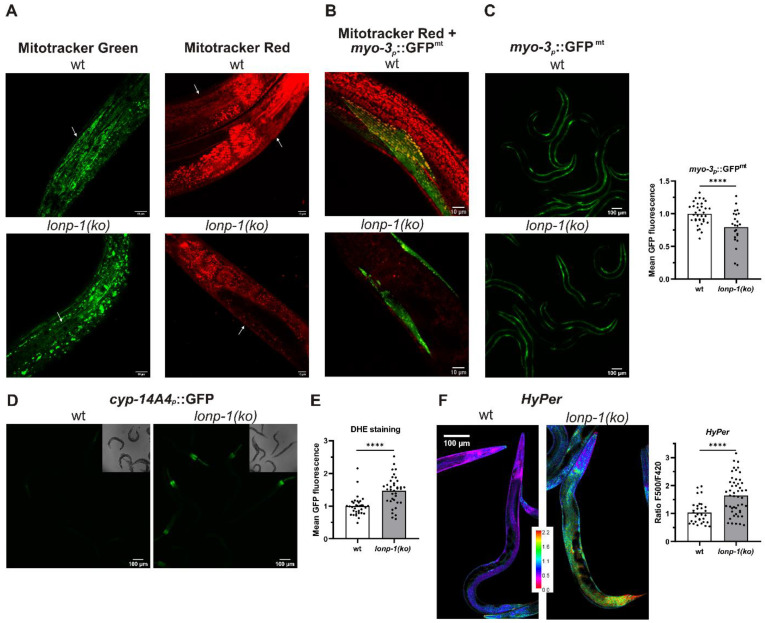
Disturbed mitochondrial network and increased ROS production in *lonp-1* mutants. (**A**) Representative confocal images of vitally stained *C. elegans* mitochondria using fluorescent dye Mitotracker Green and Mitotracker Red CMXRos in N2 (wt) and *lonp-1(ko)* animals. Scale bar, 10 μm. (**B**) Representative confocal images of the *myo-3_p_::GFP^mt^* transgenic animals that express GFP localized to muscle mitochondria, stained with Mitotracker Red CMXRos. Scale bar, 10 μm. (**C**) Representative microscopy images and quantification of fluorescence of the *myo-3_p_::GFP^mt^* reporter in wt and *lonp-1(ko)* mutants. Scale bar, 100 μm. (**D**) Representative microscopy images of transgenic animals expressing a *cyp-14A4**p::gfp* fusion gene that is induced by mitochondrial dysfunction in wt and *lonp-1(ko)* mutants. Scale bar, 100 μm. (**E**) Quantification of fluorescence in wt and *lonp-1(ko)* animals stained with the ROS-sensitive dye dihydroethidium (DHE). (**F**) Representative confocal images and quantification of the ratio of oxidized to reduced ratiometric reporter HyPer in wt and *lonp-1(ko)* worms expressing jrIs1[*rpl-17p::HyPer*]. Scale bar, 100 μm. All experiments were performed on 1-day adults, and an unpaired *t*-test was used to assess significance (**** *p* value < 0.0001).

**Figure 3 cells-11-01363-f003:**
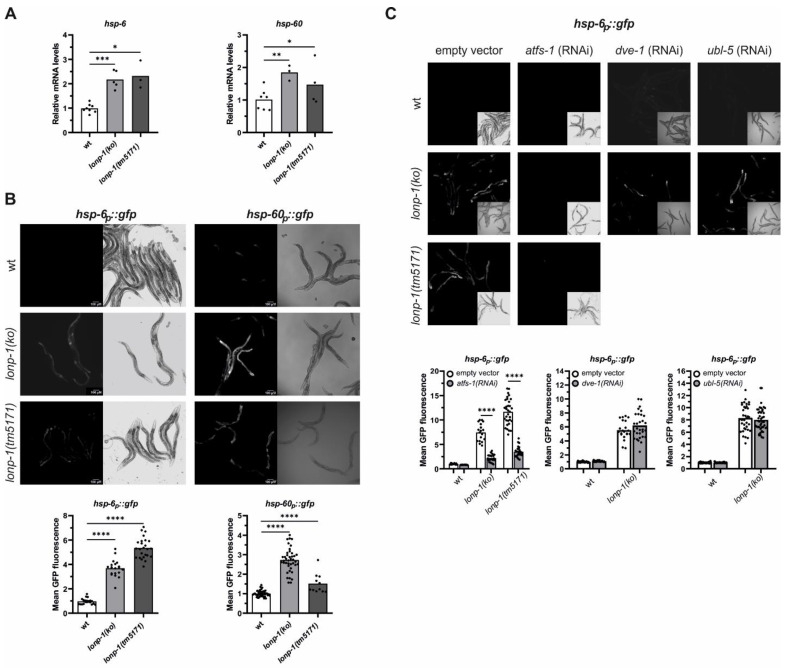
Deletion of *lonp-1* induces UPR^mt^ in *C. elegans*. (**A**) Quantification of the relative mRNA levels of endogenous *hsp-6* and *hsp-60* genes in wt and *lonp-1* worms on the first day of adulthood. The normalized mean fold-change of all biological replicates relative to control strain is shown, and significance was assessed by paired *t*-test. (**B**) Representative microscopy images and GFP fluorescence quantification of the UPR^mt^ reporters *hsp-6p::gfp* and *hsp-60p::gfp* in wt and *lonp-1* adults. In both graphs, an unpaired *t*-test was used to assess significance (*p* value). (**C**) Representative microscopy images and GFP quantification of the UPR^mt^ reporter *hsp-6p::gfp* in 1-day adult wt and *lonp-1* mutants, subjected throughout their life to RNAi against *atfs-1*, *dve-1,* or *ubl-5*, compared with animals fed the empty vector. Two-way ANOVA followed by post hoc Tukey’s test was used to assess the significance of treatment in each strain (showed as asterisks in each graph) and the interaction between genotype and RNAi treatment (*p* < 0.0001 for *atfs-1(RNAi)* in both *lonp-1* mutants, *p* = 0.2510 for *dve-1(RNAi)* and *p* = 0.4728 for *ubl-5(RNAi)* in *lonp-1(ko)* worms). In all panels, asterisks denote statistical significance: * *p* = 0.01–0.05, ** *p* = 0.001–0.01, *** *p* = 0.0001–0.001, **** *p* < 0.0001.

**Figure 4 cells-11-01363-f004:**
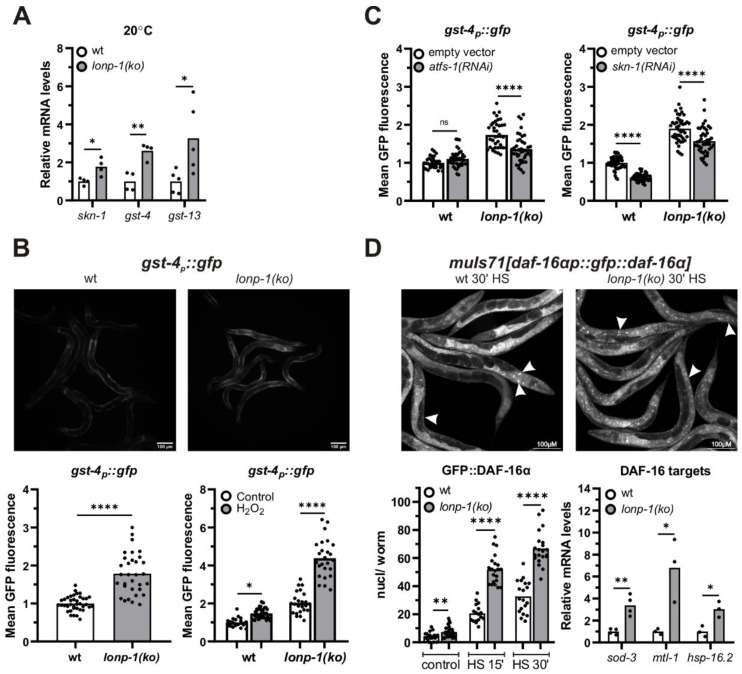
Activation of antioxidant and stress response mechanisms in *lonp-1* mutants. (**A**) Quantification of the relative mRNA levels of endogenous *skn-1* and its target genes, *gst-4* and *gst-13*, in 1-day adult wt and *lonp-1(ko)* worms grown at 20 °C. The normalized mean fold-change of all biological replicates relative to control strain is shown, and significance was assessed by paired *t*-test. (**B**) Representative microscopy images of *gst-4p::gfp* reporter in wt and *lonp-1(ko)* 1-day adults, under normal conditions and GFP fluorescence quantification in these animals as well as following exposure to ROS-generator H_2_O_2_ (10 mM for 30 min, followed by 30 min recovery before visualization). To assess significance, an unpaired *t*-test was used for worms under normal conditions, while two-way ANOVA followed by post hoc Tukey’s test was used to assess the significance of H_2_O_2_ treatment in each strain (showed as asterisks in the graph) and the interaction between genotype and treatment (*p* < 0.0001). (**C**) GFP quantification of the *gst-4p::gfp* reporter in wt and *lonp-1(ko)* 1-day adults subjected from eggs to RNAi against *atfs-1* or *skn-1.* Two-way ANOVA followed by post hoc Tukey’s test was used to assess the significance of treatment in each strain (showed as asterisks in graphs) and the interaction between genotype and each RNAi (*p* < 0.0001 for *atfs-1(RNAi)* and *p* = 0.4774 for *skn-1(RNAi)* treatment). (**D**) Representative microscopy images and quantification of fluorescent nuclei in wt and *lonp-1(ko)* young adults expressing the *muIs71[daf-16ap::gfp::daf-16a(bKO)]* translational reporter, under normal or mild heat-shock conditions (HS for 15 or 30 min at 35 °C), with white arrows to indicate localization of the reporter in intestinal and epidermal nuclei. In the right graph, quantification of the relative mRNA levels of endogenous DAF-16 target genes in 1-day adult wt and *lonp-1(ko)* worms grown at 20 °C is shown. The normalized mean fold-change of all biological replicates is shown, and significance was assessed by paired *t*-test. In all panels, asterisks denote statistical significance: * *p* = 0.01–0.05, ** *p* = 0.001–0.01, *** *p* = 0.0001–0.001, **** *p* < 0.0001.

**Figure 5 cells-11-01363-f005:**
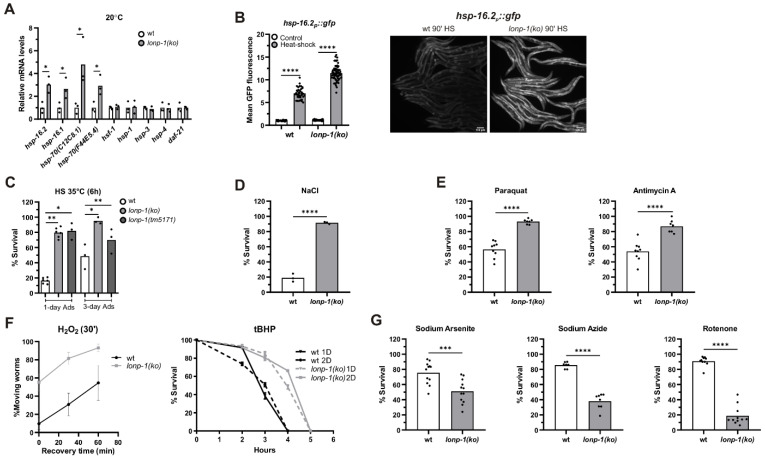
Loss of *lonp-1* enhances heat shock response (HSR) and induces diverse organismal responses to various stresses. (**A**) Quantification of the relative mRNA levels of endogenous HSR genes, and UPR^ER^-related genes, in *lonp-1* worms under normal growth temperature (20 °C). The normalized mean fold-change of all biological replicates relative to control strain is shown, and significance was assessed by paired *t*-test. (**B**) Representative microscopy images and GFP fluorescence quantification of the *hsp-16.2_p_**::gfp* reporter in wt and *lonp-1(ko)* worms subjected to heat shock (HS at 35 °C for 90 min). Two-way ANOVA followed by post hoc Tukey’s test was used to assess the significance of treatment in each strain (showed as asterisks in the graph) and the interaction between genotype and HS (*p* < 0.0001). Scale bar, 100 μm. Resistance to **(C)** HS (35 °C for 6 h), (**D**) osmotic stress (500 mM NaCl for 24 h), (**E**) paraquat (30 mM for 48 h) or antimycin A (40 μm for 24 h) and (**F**) H_2_O_2_ (10 mM for 30 min) or tBHP (10 mM), of *lonp-1* mutants at day 1 or 2 of adulthood, at the indicated time periods. (**G**) On the contrary, *lonp-1* mutants exhibit increased sensitivity to oxidants sodium arsenite (7.5 mM), sodium azide (1.5 mM), and rotenone (10 μm) after 24 h exposure on day 1 of adulthood. The percentage survival for all biological replicates was plotted, and an unpaired *t*-test was used to assess significance (*p* value). In all panels, asterisks denote statistical significance: * *p* = 0.01–0.05, ** *p* = 0.001–0.01, *** *p* = 0.0001–0.001, **** *p* < 0.0001.

**Figure 6 cells-11-01363-f006:**
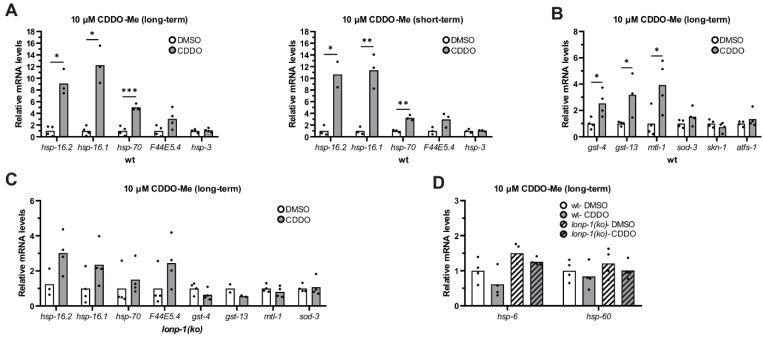
CCDO-Me induces specific stress-responsive genes in wt worms. (**A**) Quantification of the relative mRNA levels of endogenous HSR genes in wt 1-day adult worms, treated with 10 μm CDDO-Me from eggs (long-term) or at the L4 stage for 24 h (short-term), under normal growth temperature (20 °C). (**B**) Quantification of the relative mRNA levels of endogenous oxidative stress genes in wt 1-day adult worms, treated with 10 μm CDDO-Me from eggs (long-term), under normal growth temperature (20 °C). (**C**) Quantification of the relative mRNA levels of all tested stress genes in *lonp-1(ko)* 1-day adults, treated with 10 μm CDDO-Me from eggs (long-term), under normal growth temperature (20 °C). (**D**) Quantification of the relative mRNA levels of endogenous UPR^mt^ genes in wt and *lonp-1(ko)* 1-day adult worms, treated with 10 μm CDDO-Me from eggs (long-term), under normal growth temperature (20 °C). In all graphs, the normalized mean fold-change of all biological replicates relative to control strain is shown, and significance was assessed by paired *t*-test (* *p* = 0.01–0.05, ** *p* = 0.001–0.01, *** *p* = 0.0001–0.001).

**Figure 7 cells-11-01363-f007:**
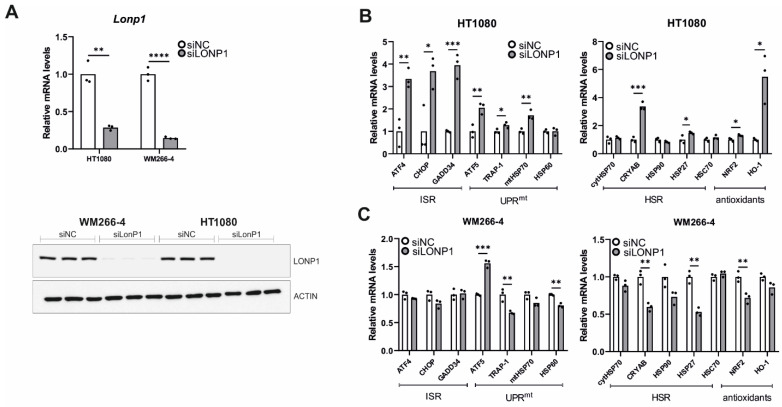
Effects of LonP1 silencing on stress-related gene expression in cancer cells. (**A**) Quantification of LonP1 relative mRNA levels and Western blot analysis presenting LonP1 versus β-ACTIN protein levels upon LonP1 silencing. Three biological replicates are represented in the blot. Quantification of relative mRNA levels of genes related to ISR (Integrated Stress Response), UPR^mt^ (mitochondrial Unfolded Protein Response), HSR (Heat Shock Response), and antioxidant responses in (**B**) fibrosarcoma (HT1080) and (**C**) metastatic melanoma (WM266-4) cell lines upon LonP1 silencing. In all graphs, the normalized mean fold-change of three biological replicates relative to the control is shown. Statistical significance was assessed by paired *t*-test. In all panels, asterisks denote statistical significance: * *p* = 0.01–0.05, ** *p* = 0.001–0.01, *** *p* = 0.0001–0.001, **** *p* < 0.0001.

**Figure 8 cells-11-01363-f008:**
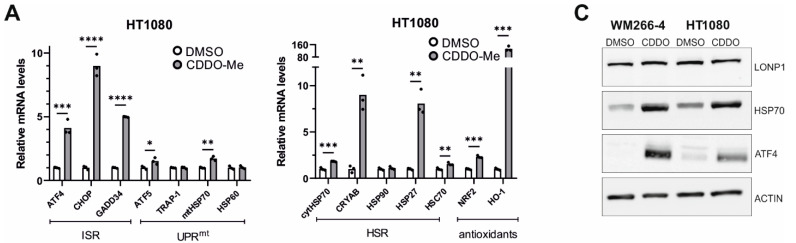
Effects of LonP1 pharmacological inhibition with CDDO-Me on stress-related gene expression. Quantification of relative mRNA levels after CDDO-Me treatment (1 μΜ, 24 h) of genes related to ISR (Integrated Stress Response), UPR^mt^ (mitochondrial Unfolded Protein Response), HSR (Heat Shock Response), and antioxidant responses in (**A**) fibrosarcoma (HT1080) and (**B**) metastatic melanoma (WM266-4) cell lines. In all graphs, the normalized mean fold-change of three biological replicates relative to the control is shown. Statistical significance was assessed by paired *t*-test (* *p* = 0.01–0.05, ** *p* = 0.001–0.01, *** *p* = 0.0001–0.001, **** *p* < 0.0001). (**C**) Western blot analysis of LonP1, HSP70, ATF4, and β-ACTIN from total protein extracts, in WM266-4 and HT1080 cancer cell lines, upon 1 μM CDDO-Me treatment for 24 h. Experiments were repeated three times, while here, one representative blot is shown.

**Figure 9 cells-11-01363-f009:**
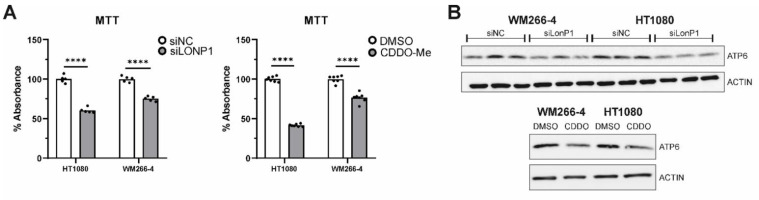
Effects of LonP1 disruption on mitochondrial function and motility in cancer cells. (**A**) MTT (3-(4,5-dimethylthiazol-2-yl)-2,5-diphenyltetrazolium bromide) assays were performed in fibrosarcoma (HT1080) and metastatic melanoma (WM266-4) cell lines after LonP1 silencing (left) or 1 μΜ CDDO-Me treatment for 24 h (right). MTT assays were carried out at least three times, whereas significance was assessed by paired *t*-test (**** *p* < 0.0001). (**B**) Western blotting of ATP6 (25 kDa) upon LonP1 silencing (above) or 1 μM CDDO-Me treatment for 24 h (below) in WM266-4 and HT1080 cancer cells. β-ACTIN was used as a protein of reference. (**C**) Scratch-wound assays were carried out for 24 h using HT1080 and WM266-4 cancer cells under control conditions versus LonP1 silencing or treatment with 500 nM of CDDO-Me. Observations were made under an inverted microscope, and pictures were taken at 4× magnification. Experiments were repeated three times, while here, one representative experiment is shown.

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
