# Peer review of "Organismal and Cellular Stress Responses upon Disruption of Mitochondrial Lonp1 Protease"

_cells, 2022, doi:10.3390/cells11081363_

Round 1

Reviewer 1 Report

Authors investigated the role of LonP1 on organismal and cellular stress response confirming previous observations regarding the impact of this protein on mitochondrial functions and stress response. Many of the findings presented are not particularly novel. However, the study is potentially interesting, but still lack some important data.

Here are main points to address:

  1. CDDO-Me may induce apoptosis and mitochondrial alterations by mechanisms other than inhibition of Lon protease. I would suggest the authors to test some more specific Lon protease inhibitors both in worms and in cancer cells.

  1. CDDO-Me is known to induce cell death. The authors used CDDO-Me at 5uM and 10 uM. At this concentrations it is toxic to cells. The authors should demonstrate, at least in cancer cells lines, that the effects they described characterize viable cells. Cell death assays, like annexin-V/propidium iodide staining or similar, should be used.

  1. The authors showed western blot for HSP70 and ATF4 in Figure 8C. What about the expression of these proteins in cells where Lon P1 was silenced by si-RNA?

  1. MTT assay is used to analyse cytotoxicity. The authors stated that “siRNA-mediated as well as pharmacological inhibition of LonP1 were found to interfere with metabolic activity rate of mitochondria as shown by the de- creased relative percentages in the MTT assays in both cell lines”. However, to measure the metabolic activity of cells, in particular oxidative phosphorylation, the Seahorse technology should be used.

Reviewer 2 Report

Eirini Taouktsi et al. report that disruption of mitochondrial LonP1 protease affects organismal and cellular stress responses. They showed the effects of CDDO-Me and LonP1 KO in the worm C elegans and human cancer cells. This study is potentially interesting. However, there are several concerns listed below.

This paper will be strengthened by addressing the following issueã„´.

  1. The authors need to have balance between C elegans data (fig1-fig6) and human cancer cells data (fig 7-fig9). Although the loss of LonP1 did affect survival of the worm C elegans and human cancer cells, LonP1 inhibition showed some different phenotype between the worm C elegans and human cancer cells. Therefore, we could not get solid conclusion.
  2. Although CDDO-Me is well known to LonP1 inhibitor, this drug has an effect on LonP1 independent mitochondrial fission (Cells 2019 8(8):833). And in human cancer cells data, the effect of CDDO-Me is a little bit different from it of Lonp1 siRNA experiment. Therefore, the authors need to add some discussion about this difference.
  1. The authors have used the Lonp1 mutant in the worm C elegans but not in human cancer cells. If possible, the authors should add the Lonp1 mutant data in human cancer cells.

Round 2

Reviewer 1 Report

Authors provide satisfying answers to the majority of issues raised during the first round of revision.